# Efficient Reinforcement Learning with Prior Causal Knowledge

**Yangyi Lu**                                           YYLU@UMICH.EDU
*University of Michigan*

**Amirhossein Meisami**                               MEISAMI@ADOBE.COM
*Adobe Inc.*

**Ambuj Tewari**                                       TEWARIA@UMICH.EDU
*University of Michigan*

Editors: Bernhard Schölkopf, Caroline Uhler and Kun Zhang

## Abstract

We introduce causal Markov Decision Processes (C-MDPs), a new formalism for sequential decision making which combines the standard MDP formulation with causal structures over state transition and reward functions. Many contemporary and emerging application areas such as digital healthcare and digital marketing can benefit from modeling with C-MDPs due to the causal mechanisms underlying the relationship between interventions and states/rewards. We propose the causal upper confidence bound value iteration (C-UCBVI) algorithm that exploits the causal structure in C-MDPs and improves the performance of standard reinforcement learning algorithms that do not take causal knowledge into account. We prove that C-UCBVI satisfies an $\tilde{O}(HS\sqrt{ZT})$ regret bound, where $T$ is the the total time steps, $H$ is the episodic horizon, and $S$ is the cardinality of the state space. Notably, our regret bound does not scale with the size of actions/interventions ($A$), but only scales with a causal graph dependent quantity $Z$ which can be exponentially smaller than $A$. By extending C-UCBVI to the factored MDP setting, we propose the causal factored UCBVI (CF-UCBVI) algorithm, which further reduces the regret exponentially in terms of $S$. Furthermore, we show that RL algorithms for linear MDP problems can also be incorporated in C-MDPs. We empirically show the benefit of our causal approaches in various settings to validate our algorithms and theoretical results.

Keywords: Reinforcement learning; Causality; Markov Decision Process (MDP).

## 1. Introduction

In reinforcement learning (RL), the agent interacts with the environment sequentially aiming to maximize its cumulative reward within a given time period. The environment is generally modeled as a Markov Decision Process (MDP) that is not fully known to the agent. At every round $t$, the agent observes the current state $s_t$ and performs an action $a_t$ according to the policy learned so far. Then the environment returns a reward $r(s_t, a_t)$ and transitions the agent to the next state $s_{t+1}$ according to the underlying state transition dynamics. The performance is usually evaluated by cumulative regret, i.e., the reward difference between the optimal policy and the agent's policy.

Many RL algorithms have been developed for the tabular setting (Jaksch et al., 2010; Bartlett and Tewari, 2012; Osband et al., 2013; Azar et al., 2017; Zhang and Ji, 2019; Wang et al., 2020; Zhang et al., 2020) where the state and action spaces have small cardinalities. Their regret or sample

complexity bounds all scale with the number of states $S$ and actions $A$ which can be very large in practice.

In healthcare applications, the doctor adjusts several features to achieve some desirable clinical outcomes (Liu et al., 2020). For example, different dose levels on medicines, types of exercises, amount of exercises, sleeping time among other conditions may affect patients' overall health condition. Patients in different states (as captured by, say, BMI, blood pressure, status of organs/body systems) usually respond differently to a given treatment. As the treatment actions are taken, the state of patients will change accordingly. In digital marketing, online advertising companies aim at attracting customers to buy products by sending marketing emails. Marketers adjust several variables such as types of products, email content, the time of day to send the email, purposes (promotion, online events, etc.) of the email, in order to improve the likelihood of a customer buying products. Customer in different dynamic states, such as loyalty levels and willingness to shop, and different intrinsic states, such as gender, age, and purchasing power, behave differently after receiving commercial emails. These examples can be modeled as MDPs where the reward variable is the overall health condition or the actual purchase. In both cases, the number of interventions and states are exponentially large.

To circumvent the curse of dimensionality where $A$ is enormous, we take a *causal approach*. In the healthcare problem, the medical and life-style treatments do not affect the state transition or reward directly but indirectly through a few *key* variables (that cannot be manipulated directly, e.g., micronutrient levels, blood oxygen level etc.) that have a direct causal effect on next states and rewards. Similarly, in the digital marketing problem, interventions on email features affect the state transition and the actual purchase (reward) through *key* variables such as interest/demand for products, price performance, engagement and whether any product has been added to the cart or not. The causal relations among manipulable variables, *key* variables and other variables in the system can be represented by a causal graph. If we have such prior causal knowledge, we do not need to treat all interventions independently as standard RL approaches do. Instead, we can connect the intervention set with the low dimensional *key* variables in order to reduce the amount of exploration. Based on the above idea, we introduce a new formalism: *causal MDPs* (C-MDPs), and prove that the regret of our algorithm *causal upper confidence bound* (C-UCBVI) no longer scales with $A$, however, it only scales with a causal graph dependent quantity $Z$. We show that there can be cases where $Z$ is exponentially smaller than $A$.

Furthermore, in order to deal with problems where *both* $S$ and $A$ are large, we propose two approaches under different assumptions on low-dimensional structures. Firstly, when the state space can be factorized as $\mathcal{S} = \mathcal{S}_1 \times \cdots \times \mathcal{S}_m$, we introduce *causal factored MDPs* (CF-MDPs). Structured relations among states can be exploited when the agent has prior understandings on the environment. For example, in the healthcare problem, we may know that at one time-step, the state of an organ is usually influenced by the states of its closely related parts, not the entire body. Combining our causal approach with factored MDP techniques, we propose causal factored UCBVI (CF-UCBVI) algorithm. We analyze its regret and prove that the explicit dependence on the state size $S$ can be eliminated. In a nutshell, we deal with large $A$ using causal relations while dealing with large $S$ with state factorizations. This approach is different from factored MDPs which directly factorize $\mathcal{S} \times \mathcal{A}$. In Section 2, we discuss the differences in more detail. We show that when $\mathcal{A}$ is exponentially large but cannot be factorized with $\mathcal{S}$, standard factored MDP approaches can fail and our causal approach

is necessary. We emphasize that 1) neither factored MDP nor our causal approach can imply one another and 2) the type of available prior knowledge on $\mathcal{S}$ and $\mathcal{A}$ should determine which method one should use. Secondly, when the state transition and reward functions can be modeled linearly with feature vectors over the state and key variable pairs, we show that RL algorithms for standard linear MDPs (Jin et al., 2019) can be incorporated in C-MDPs.

**Our Contributions.** We summarize our main contributions below:

1. We study a new formalism: causal MDPs, in which we search for good interventions over an exponentially large space. In the bandit literature, *causal bandits* have been studied recently (Lattimore et al., 2016; Sen et al., 2017; Lu et al., 2019; Nair et al., 2020) where researchers have used causal graphs to model the relations among interventions and the reward. *In this paper, we extend the idea behind causal bandits to MDPs.* We propose causal upper confidence bound (C-UCBVI) algorithm that enjoys $\tilde{O}(HS\sqrt{ZT})$ regret. In our regret bound, $Z$ is a causal graph dependent quantity that can be exponentially smaller than the number of actions $A$. Our result is superior to the guarantees available for standard RL algorithms whose regret scales with $A$.

2. Building on causal MDPs, we propose two approaches to deal with cases when the state space is also enormous. In our first approach, we introduce causal factored MDPs. We propose causal factored upper confidence bound (CF-UCBVI) algorithm that achieves $\tilde{O}(H \sum_{i=1}^{m} \sqrt{S_i S[I_i] ZT})$ regret when we factorize $\mathcal{S}$ as $\mathcal{S}_1 \times \cdots \times \mathcal{S}_m$. In this result, $S_i$ and $\mathcal{S}[I_i]$ denotes the cardinalities for $\mathcal{S}_i$ and $\mathcal{S}$ restricted to scope $I_i$ [1], which can both be exponentially smaller than the number of states $S$. In the second approach, we show that existing linear MDP algorithm can be well adapted to causal MDP problems and achieve $\tilde{O}(\sqrt{d^3 H^3 T})$ regret, where $d$ is the dimension for features over the state and *key* variable pairs. Both approaches reduce $S$ dependency from the regret.

Our key idea is that we use prior causal knowledge such as causal graphs to obtain conditional independence relations among action, reward and state variables and use them to develop efficient algorithms.

## 2. Related work

Our work on causal (factored) MDPs is directly inspired by recent work on *causal bandit* problems (Bareinboim et al., 2015; Lattimore et al., 2016; Sen et al., 2017; Lee and Bareinboim, 2018; Lu et al., 2019; Nair et al., 2020), where the arms of the bandit problem are interventions on a set of variables and their relations with the reward are captured by a causal graph (Pearl, 2000). In causal bandits, the causal graph is composed of manipulable/non-manipulable variables and the reward variable. For causal (factored) MDPs, we need to consider two types of graphs: one is the reward graph and the other is the state transition graph. Our proposed causal MDP algorithms that exploit these two types of causal graphs in order to learn the MDP dynamics efficiently.

There is another line of work on causal reinforcement learning (Zhang and Bareinboim, 2016, 2019; Namkoong et al., 2020; Zhang, 2020; Wang et al., 2021) studying MDPs or dynamic treatment regimes with unobserved confounders. Our paper does not focus on confounding issues. We model the related variables by causal graphs following the idea behind *causal bandits*.

---

1. We provide formal definitions in Section 3.

A classic approach to deal with exponentially large state and action spaces is to use factored MDPs. Recent work has provided formal regret guarantees for factored MDPs (Osband and Van Roy, 2014; Xu and Tewari, 2020; Tian et al., 2020). The key idea is to factorize the state set ($\mathcal{S}$) and state-action set ($\mathcal{S} \times \mathcal{A}$): $\mathcal{S} = \mathcal{S}_1 \times \cdots \times \mathcal{S}_m$, $\mathcal{S} \times \mathcal{A} = \mathcal{X}_1 \times \cdots \times \mathcal{X}_n$. The state transition and reward dynamics are generated based on these two factorization structures. However, in practice, actions do not always have local effect on the outcomes and thus cannot always be factorized together with states as described above. For example, it is almost impossible to directly factorize the Cartesian product between the state of organs/BMI/etc. and the treatments (which serves as actions), because some medical treatments, especially life-style treatments usually affect all of the organs/BMI/etc. Same for email campaign, all the email features affect the loyalty building and the actual purchase to some degree. In these cases, the state-action set can only be written as: $\mathcal{S} \times \mathcal{A} = \mathcal{S}_1 \times \cdots \times \mathcal{S}_m \times \mathcal{A}$. Under this condition, existing factored-MDP algorithms cannot avoid a dependence on $A$ in their regret, so our causal approaches are necessary. We emphasize that our causal approaches and factored MDP approaches have their advantages under different assumptions. Causal approaches are preferred when there is prior causal knowledge on $\mathcal{S}, \mathcal{A}$ and the outcomes, while factored MDP approaches are preferred when actions have local effect on the outcomes so that $\mathcal{A}$ can be factorized with $\mathcal{S}$.

Function approximation methods (Jin et al., 2019; Yang and Wang, 2020; Zhou et al., 2020; Ayoub et al., 2020) are also powerful tools to handle large state and action spaces when certain feature maps are available. Our methods instead exploit a different type of side information: the underlying causal graph, to address problems with large $S$ and $A$. To reduce sample complexity, dynamic Bayesian networks (DBNs) are also used to model state transitions for every action (Boutilier et al., 2000), where an action may lead to sparse connections among state variables at consecutive time steps. In contrast, in our setting each manipulable variable at time $t$ will connect with all other manipulable variables at time $t + 1$. We do have conditional independence between actions and the reward, actions and state transitions at each time step, but not across different time steps. Outside of RL, similar ideas of causal modeling in time-dependent systems have also been studied (Blondel et al., 2017; Srinivasan et al., 2021).

## 3. Preliminaries

We follow standard RL/graphical terminology and notation in Azar et al. (2017) and Koller and Friedman (2009) to state the casual (factored) MDP problems.

**Causal Graph.** A directed acyclic graph $\mathcal{G}$ is used to model the causal structure over a set of random variables $\mathbf{X} = \{X_1, \ldots, X_N\}$. We write the domain for variable $X$ as $\mathrm{Dom}(X)$ and the joint distribution over $\mathbf{X}$ along graph $\mathcal{G}$ at state $s$ as $P(\cdot|s)$. The parents of a variable $X_i$, denoted by $\mathrm{Pa}_{X_i}$, include all variables $X_j$ such that there is an edge from $X_j$ to $X_i$ in $\mathcal{G}$. A size $m$ intervention (action) corresponds to $\mathrm{do}(\mathbf{X}_{\mathrm{sub}} = \mathbf{x})$ such that $|\mathbf{X}_{\mathrm{sub}}| = m$, which assigns the values $\mathbf{x} = \{x_1, \ldots, x_m\}$ to the corresponding variables. For each variable $X \in \mathbf{X}_{\mathrm{sub}}$, the intervention also removes all edges from $\mathrm{Pa}_X$ to $X$ and the resulting graph defines a probability distribution $P(\mathbf{X}_{\mathrm{sub}}^c|s, \mathrm{do}(\mathbf{X}_{\mathrm{sub}} = \mathbf{x}))$ over $\mathbf{X}_{\mathrm{sub}}^c := \mathbf{X} \setminus \mathbf{X}_{\mathrm{sub}}$. We use $|\cdot|$ to denote the cardinality of a set.

**MDP.** A tabular episodic MDP is defined by a tuple $(\mathcal{S}, \mathcal{A}, \mathbb{P}, R, H)$, where $\mathcal{S}$ and $\mathcal{A}$ are the set of states and actions with cardinalities $|\mathcal{S}| = S$ and $|\mathcal{A}| = A$, $H$ is the planning horizon in each episode, $\mathbb{P}$ is the state transition matrix such that $\mathbb{P}(\cdot|s, a)$ gives the distribution over next state if an action $a$ is taken on state $s$, and $R : \mathcal{S} \times \mathcal{A} \to [0, 1]$ is the deterministic reward function over a state

action pair. The agent interacts with the environment in a sequence of episodes: an initial state $s_1$ is picked arbitrarily by an adversary. At each step $h \in [H]$, the agent observes state $s_h \in \mathcal{S}$, picks an action $a_h \in \mathcal{A}$ and receives reward $R(s_h, a_h)$. The episode ends when $s_{H+1}$ is reached.

The policy is expressed as a mapping $\pi : \mathcal{S} \times [H] \to \mathcal{A}$. We use $V_h^\pi : \mathcal{S} \to \mathbb{R}$ to denote the value function at step $h$ under policy $\pi$, so that $V_h^\pi(s)$ gives the expected sum of remaining rewards received under policy $\pi$, starting from $s_h = s$, until the end of episode:

$$V_h^\pi(s) \overset{\text{def}}{=} \mathbb{E}\left[\sum_{h'=h}^{H} R(s_{h'}, \pi(s_{h'}, h')) | s_h = s\right].$$

We use $Q_h^\pi : \mathcal{S} \times \mathcal{A} \to \mathbb{R}$ to denote $Q$-value function at step $h$ under policy $\pi$ so that $Q_h^\pi(s, a)$ gives the expected sum of remaining rewards received under policy $\pi$, starting from $s_h = s, a_h = a$, till the end of the episode:

$$Q_h^\pi(s, a) \overset{\text{def}}{=} R(s, a) + \mathbb{E}\left[\sum_{h'=h+1}^{H} R(s_{h'}, \pi(s_{h'}, h')) \mid s_h = s, a_h = a\right].$$

An optimal policy $\pi^*$ gives the optimal value $V_h^*(s) := \sup_\pi V_h^\pi(s)$ for all $s \in \mathcal{S}$ and $h \in [H]$. The policy $\pi$ at every step $h$ defines the state transition kernel $\mathbb{P}_h^\pi$ and the reward function $r_h^\pi$ as $\mathbb{P}_h^\pi(y|s) \overset{\text{def}}{=} \mathbb{P}(y|s, \pi(s, h))$ and $r_h^\pi(s) \overset{\text{def}}{=} R(s, \pi(s, h))$ for all $s$. For every $V : \mathcal{S} \to \mathbb{R}$ the right linear operators $\mathbb{P}\cdot$ and $\mathbb{P}_h^\pi\cdot$ are also defined as $(\mathbb{P}V)(s, a) \overset{\text{def}}{=} \sum_{s' \in \mathcal{S}} \mathbb{P}(s'|s, a)V(s')$ for all $(s, a)$ and $(\mathbb{P}_h^\pi V)(s) \overset{\text{def}}{=} \sum_{s' \in \mathcal{S}} \mathbb{P}_h^\pi(s'|s)V(s')$ for all $s$, respectively.

**Causal MDP (C-MDP).** In causal MDPs, the actions are composed by interventions. At every state $s$, we define two causal graphs: the reward graph $\mathcal{G}^R(s)$ and the state transition graph $\mathcal{G}^S(s)$. We denote the reward and state variable by $\mathbf{R}$ and $\mathbf{S}$. The learner can intervene on variables $\mathbf{X}^I$, while the parent variables of $\mathbf{R}$: $\mathbf{Z}^{\mathbf{R}} := \text{Pa}_{\mathbf{R}}$ and the parent variables of $\mathbf{S}$: $\mathbf{Z}^{\mathbf{S}} := \text{Pa}_{\mathbf{S}}$ cannot be intervened. [2] Precisely, the action (intervention) set is:

$$\mathcal{A} = \{\text{do}(\mathbf{X}_{\text{sub}} = \mathbf{x}) \mid \mathbf{X}_{\text{sub}} \in \mathbf{X}^I, \mathbf{x} \in \text{Dom}(\mathbf{X}_{\text{sub}})\}.$$

At every state $s$, causal graphs $\mathcal{G}^{\mathbf{R}}(s)$ and $\mathcal{G}^{\mathbf{S}}(s)$ contain variables $\mathbf{X}^{\mathbf{R}} = \mathbf{X}^I \cup \mathbf{Z}^{\mathbf{R}} \cup \mathbf{R}$ and $\mathbf{X}^{\mathbf{S}} = \mathbf{X}^I \cup \mathbf{Z}^{\mathbf{S}} \cup \mathbf{S}$, respectively. Note that the identity of variables on causal graphs does not vary by state, but the underlying distributions can change. In Figure 1, we use a digital marketing example to explain these notations.

In our causal MDP algorithms, a learner is given the intervention set $\mathcal{A}$, the identity of parent variables $\mathbf{Z} := \mathbf{Z}^{\mathbf{R}} \cup \mathbf{Z}^{\mathbf{S}}$ and conditional distributions of $\mathbf{z} \in \mathcal{Z}$ given a $(s, a) \in \mathcal{S} \times \mathcal{A}$ pair: $P(\mathbf{z}|s, a)$, where $\mathcal{Z}$ denotes the domain set for $\mathbf{Z}$. We use $Z$ as the size of $\mathcal{Z}$. At each step $h \in [H]$, the learner observes a reward $R(s_h, a_h)$ and the realizations of $\mathbf{Z}$: $\mathbf{z}_h$. Using these causal

---

2. Otherwise, one can simply restrict the intervention set to those only intervening over $\text{Pa}_{\mathbf{R}} \cup \text{Pa}_{\mathbf{S}}$, then the problem is trivially reduced to a standard MDP problem.

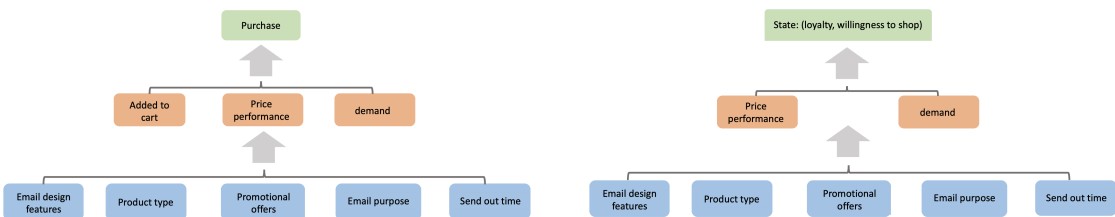

Figure 1: Reward causal graph (left) and state transition causal graph (right) at the current state $s_t$ for the digital marketing problem. In each graph, all variables can be categorized into three layers: the outcome variable on top layer corresponds to the green node, i.e. the reward variable ($\mathbf{R}$) and the state variable ($\mathbf{S}$); the manipulable variables regarding to emails on bottom layer are in blue ($\mathbf{X}^I$); direct parent variables of the outcome variables ($\mathbf{Z^R}$ and $\mathbf{Z^S}$) are marked in orange between the green and blue. We use grey arrows to describe the complex causal relationships between any connected two layers. At every time $t$, marketers adjust the blue nodes and passively observe the values of nodes in orange and green afterwards. Other than $P(\mathbf{z}|s,a)$ quantities, our causal MDP algorithms only require the knowledge of the identity of manipulable variables, the outcome variables and their corresponding direct parents, instead of the entire causal structures including all graph edges.

information, one can re-write the state-transition and reward functions as follows:

$$\mathbb{P}(s'|s,a) = \sum_{\mathbf{z}\in\mathcal{Z}} \mathbb{P}(s'|s,\mathbf{Z}=\mathbf{z})P(\mathbf{Z}=\mathbf{z}|s,a),$$

$$R(s,a) = \sum_{\mathbf{z}\in\mathcal{Z}} R(s,\mathbf{Z}=\mathbf{z})P(\mathbf{Z}=\mathbf{z}|s,a),$$

where $R(s,\mathbf{Z}=\mathbf{z}) \stackrel{\text{def}}{=} \mathbb{E}[\mathbf{R}|s,\mathbf{z}]$ denotes the expected reward given a state and parent pair. We next define a q-value function: $q_h^\pi : \mathcal{S} \times \mathcal{Z} \to \mathbb{R}$, such that $q_h^\pi(s,\mathbf{z})$ gives the expected sum of rewards received under policy $\pi$, starting from $s_h = s, \mathbf{z}_h = \mathbf{z}$, till the end of the episode:

$$q_h^\pi(s,\mathbf{z}) \stackrel{\text{def}}{=} R(s,\mathbf{z}) + \mathbb{E}\left[ \sum_{h'=h+1}^{H} R(s_{h'},\pi(s_{h'},h')) \mid s_h = s, \mathbf{z}_h = \mathbf{z} \right].$$

By definition, $Q_h^\pi(s,a)$ can be written as $\sum_{\mathbf{z}\in\mathcal{Z}} P(\mathbf{Z}=\mathbf{z}|s,a)q_h^\pi(s,\mathbf{z})$. A causal MDP is then defined as an MDP equipped with dynamic causal graphs $\mathcal{G}^\mathbf{R}, \mathcal{G}^\mathbf{S}$ and can be represented by a tuple $\mathcal{M}_C = \left(\mathcal{S}, \mathcal{A}, \mathbb{P}, R, H, \mathcal{G}^\mathbf{R}, \mathcal{G}^\mathbf{S}\right)$.

**Causal Factored MDP (CF-MDP).** A causal factored MDP is a causal MDP whose reward and state-transition dynamics have some conditional independence structures. To formally describe this problem, we first present some related factored MDP definitions.

**Definition 1 (Scope operation for factored set $\mathcal{S} = \mathcal{S}_1 \times \ldots \times \mathcal{S}_m$)** *For any subset of indices $I \subseteq \{1,\ldots,m\}$, define the scope set $\mathcal{S}[I] \triangleq \bigotimes_{i\in I} \mathcal{S}_i$. For any $s \in \mathcal{S}$, define the scope variable $s[I] \in \mathcal{S}[I]$ to be the value of the variables $s_i \in \mathcal{S}_i$ with indices $i \in I$. For singleton sets $I$, we write $s[\{i\}]$ as $s[i]$ for simplicity.*

We use $\mathcal{P}_{\mathcal{X},\mathcal{Y}}$ as a set of functions mapping elements of a finite set $\mathcal{X}$ to probability mass functions over a finite set $\mathcal{Y}$.

**Definition 2 (Factored state transition in CF-MDPs)** *The transition function class $\mathcal{P}$ is factored over $\mathcal{S} \times \mathcal{Z} = \mathcal{S}_1 \times \cdots \times \mathcal{S}_m \times \mathcal{Z}$ and $\mathcal{S} = \mathcal{S}_1 \times \cdots \times \mathcal{S}_m$ with scopes $I_1, \ldots, I_m$ if and only if, for all $\mathbb{P} \in \mathcal{P}, s, s' \in \mathcal{S}, \mathbf{z} \in \mathcal{Z}$, there exist some $\{\mathbb{P}_i \in \mathcal{P}_{\mathcal{S}[I_i] \times \mathcal{Z}, \mathcal{S}_i}\}_{i=1}^m$ such that $\mathbb{P}(s'|s, \mathbf{z}) = \prod_{i=1}^m \mathbb{P}_i(s'[i]|s[I_i], \mathbf{z})$.*

**Definition 3 (Factored deterministic reward functions)** *The reward function $R$ is factored over $\mathcal{S} \times \mathcal{Z} = \mathcal{S}_1 \times \cdots \times \mathcal{S}_m \times \mathcal{Z}$ with scopes $J_1, \ldots, J_m$, i.e. for all $s \in \mathcal{S}, \mathbf{z} \in \mathcal{Z}$, we have $R(s, \mathbf{z}) = \sum_{i=1}^m R_i(s[J_i], \mathbf{z})$.*

A causal factored MDP is then defined to be a causal MDP with factored rewards and factored transitions. We can write it as a tuple $\mathcal{M}_{CF} = \left(\{\mathcal{S}_i\}_{i=1}^m, \mathcal{A}, \{I_i\}_{i=1}^m, \{\mathbb{P}_i\}_{i=1}^m, \{J_i\}_{i=1}^m, \{R_i\}_{i=1}^m, H, \mathcal{G}^{\mathbf{R}}, \mathcal{G}^{\mathbf{S}}\right)$. Notably, we do not factorize $\mathcal{Z}$ in the state transition and reward function classes.

**Regret.** We denote the number of episodes by $K$, starting state and policy by $s_{k,1}$ and $\pi_k$ for each episode. We measure the performance of the learner over $T = KH$ steps by the total expected regret:

$$R_K := \sum_{k=1}^K (V_1^*(s_{k,1}) - V_1^{\pi_k}(s_{k,1})).$$

The goal of learner is to follow a sequence of policies $\pi_1, \ldots, \pi_K$ that minimizes $R_K$.

In this paper, we focus on the setting where the reward functions $R$ and $\{R_i\}_{i=1}^m$ are known. This assumption is just used for simplicity, because reward estimation is not the main difficulty of RL problems (Azar et al., 2017; Liao et al., 2021).

**Assumption 1 (Causal (Factored) MDP Regularity)** *For C-MDPs, we assume $\mathcal{S}, \mathcal{A}, \mathcal{Z}$ are finite sets with cardinalities $S$ and $A$ and $Z$, respectively. The immediate rewards $R(s, \mathbf{z}) \triangleq \mathbb{E}[\mathbf{R}|s, \mathbf{z}] \in [0, 1]$ are known for $s \in \mathcal{S}, \mathbf{z} \in \mathcal{Z}$. For CF-MDPs, $\mathcal{S}[I_i]$ and $\mathcal{S}[J_i]$ are finite sets with cardinalities $S[I_i]$ and $S[J_i]$. The immediate rewards in every reward scope $R_i(s[J_i], \mathbf{z}) \in [0, 1]$ are known for $s[J_i] \in \mathcal{S}[J_i], \mathbf{z} \in \mathcal{Z}, i = 1, \ldots, m$.*

## 4. Causal UCBVI

In this section, we propose and analyze an efficient algorithm for causal MDPs. We generalize *upper confidence bound value iteration* (Azar et al., 2017) algorithm (UCBVI) to its causal counterpart and show that the regret bound of our causal algorithm only scale with a factor which can be exponentially smaller than the size of interventions.

UCBVI is near-optimal when causal information is non-available. Azar et al. (2017) showed that under conditions $T \geq H^3 S^3 A$ and $SA \geq H$, using a Hoeffding "exploration bonus", one can achieve a high probability regret bound of $\tilde{O}(H\sqrt{SAT})$ while using a Bernstein-Freedman "exploration bonus", one can further achieve a minimax regret $\tilde{O}(\sqrt{HSAT})$, that matches the established lower bound $\Omega(\sqrt{HSAT})$ of (Jaksch et al., 2010) up to logarithmic factors. However, in the causal MDP setting, the intervention set is huge that makes UCBVI and other standard RL algorithms impractical since their regret all scale with $\sqrt{A}$.

---

**Algorithm 1** C-UCBVI

1: **Input:** action set $\mathcal{A}$, states $\mathcal{S}$, identity of parent variables $\mathbf{Z}$, $P(\mathbf{z}|s,a)$ terms.
2: Initialize data $\mathcal{H} = \phi$, $Q_{k,h}(s,a) = H$ for all $k,h,s,a$.
3: **for** episode $k = 1,\ldots,K$ **do**
4:     **for** step $h = 1,\ldots,H$ **do**
5:         Take action $a_{k,h} = \mathrm{argmax}_{a \in \mathcal{A}} Q_{k,h}(s,a)$ and observe values of parent variables $\mathbf{z}_{k,h}$.
6:         Update $\mathcal{H} = \mathcal{H} \cup (s_{k,h}, a_{k,h}, \mathbf{z}_{k,h}, s_{k,h+1})$.
7:     **end for**
8:     $Q_{k,h}(s,a) = $ C-UCB-Q-values$(\mathcal{H})$.
9: **end for**

---

To overcome this issue, we propose causal UCBVI (C-UCBVI) in Algorithm 1. At every episode $k$, C-UCBVI calls Algorithm 2 to update the state transition probabilities $\mathbb{P}(s'|s,\mathbf{z})$ by the frequencies of corresponding state-$\mathbf{Z}$-state and state-$\mathbf{Z}$ tuples using past data. We then follow the idea of UCBVI that updates the upper bounds of value functions and $Q$ functions at every level $h$ by value iteration using an empirical Bellman operator and a confidence bonus. However, instead of directly updating the upper confidence bound of $Q$ functions over state-action pairs, our algorithm updates the upper bounds of $q$-value functions using Hoeffding "exploration bonus" (Algorithm 3) over state-$\mathbf{Z}$ pairs denoted by $q_{k,h}(s,\mathbf{z})$. We then update the upper confidence bound of $Q$ function for every $(s,a)$ pair as following:

$$Q_{k,h}(s,a) = \sum_{\mathbf{z} \in \mathcal{Z}} P(\mathbf{z}|s,a) q_{k,h}(s,\mathbf{z}).$$

Upper bound for value functions are then updated by $V_{k,h}(s) = \max_{a \in \mathcal{A}} Q_{k,h}(s,a)$.

Given these estimated value functions, the learner at state $s$ performs the action that maximizes $Q_{k,h}(s,a)$ among all $a \in \mathcal{A}$. The environment reveals the values of variables $\mathbf{Z}$ denoted by $\mathbf{z}_{k,h}$. In summary, C-UCBVI only estimates the state transition probabilities and $q$-value functions for all $(s,\mathbf{z}) \in S \times \mathcal{Z}$ pairs.

We present our theoretical results for C-UCBVI in Theorem 4.

**Theorem 4** *With probability $\geq 1 - \delta$, the regret of C-UCBVI (Algorithm 1) is bounded by:*

$$R_K = \tilde{O}\left(HS\sqrt{ZT}\right). \tag{1}$$

*We omit small order terms that do not depend on $T = KH$.*

Above result shows that the regret of C-UCBVI does not scale with $\sqrt{A}$, instead scales with: $\sqrt{Z}$. Suppose $\mathbf{X}^I$ and $\mathbf{Z}$ contain $N$ and $n$ variables, respectively, and for simplicity we assume every variable in $\mathbf{X}^I \cup \mathbf{Z}$ can take on $k$ different values. In practical applications, $N$ is usually greater than $n$, for example, in digital marketing, the number of email features can be a lot more than the number of *key* variables such as price performance and demand. In this case, $Z = k^n$ is exponentially smaller than $A = k^N$. In summary, C-UCBVI outperforms standard RL algorithms as long as $Z \leq A$. The whole proof is in Section A.

---

**Algorithm 2** C-UCB-Q-values

---

1: **Input:** Bonus algorithm (Algorithm 3), Data $\mathcal{H}$.
2: $N_k(s, \mathbf{z}, y) = \sum_{(s', \mathbf{z}', y') \in \mathcal{H}} \mathbb{1}(s' = s, \mathbf{z}' = \mathbf{z}, y' = y)$ for all $(s, \mathbf{z}, y) \in \mathcal{S} \times \mathcal{Z} \times \mathcal{S}$.
3: $N_k(s, \mathbf{z}) = \sum_{y \in \mathcal{S}} N_k(s, \mathbf{z}, y)$ for all $(s, \mathbf{z}) \in \mathcal{S} \times \mathcal{Z}$.
4: Let $\mathcal{K} = \{(s, \mathbf{z}) \in \mathcal{S} \times \mathcal{Z}, N_k(x, \mathbf{z}) > 0\}$.
5: Estimate $\hat{\mathbb{P}}_k(y|s, \mathbf{z}) = \frac{N_k(s, \mathbf{z}, y)}{N_k(s, \mathbf{z})}$ for all $(s, \mathbf{z}) \in \mathcal{K}$.
6: Initialize $V_{k, H+1}(s) = 0$ for all $s \in \mathcal{S}$.
7: **for** $h = H, H-1, \ldots, 1$ **do**
8:    **for** $(s, \mathbf{z}) \in \mathcal{S} \times \mathcal{Z}$ **do**
9:       **if** $(s, \mathbf{z}) \in \mathcal{K}$ **then**
10:          $b_{k,h}(s, \mathbf{z}) = \text{bonus}(N_k(s, \mathbf{z}))$
11:          $q_{k,h}(s, \mathbf{z}) = \min(H, R(s, \mathbf{z}) + \hat{\mathbb{P}}_k V_{k, h+1}(s, \mathbf{z}) + b_{k,h}(s, \mathbf{z}))$
12:       **else**
13:          $q_{k,h}(s, \mathbf{z}) = H$
14:       **end if**
15:    **end for**
16:    **for** $(s, a) \in \mathcal{S} \times \mathcal{A}$ **do**
17:       $Q_{k,h}(s, a) = \sum_{\mathbf{z} \in \mathcal{Z}} P(\text{Pa}_R = \mathbf{z}|s, a) q_{k,h}(s, \mathbf{z})$
18:    **end for**
19:    $V_{k,h}(s) = \max_{a \in \mathcal{A}} Q_{k,h}(s, a)$
20: **end for**
21: **Output:** Q-values $Q_{k,h}(s, a)$ for all $(s, a) \in \mathcal{S} \times \mathcal{A}$.

---

**Algorithm 3** Bonus for C-UCBVI

---

1: **Input:** $\delta > 0, N_k(s, \mathbf{z}), L = \log(5SHKZT/\delta)$.
2: $b = 7HL\sqrt{\frac{S}{N_k(s, \mathbf{z})}}$.
3: **Output:** $b$.

---

## 5. Causal factored-UCBVI

In this section, we study causal factored MDPs, where states $s \in \mathcal{S} \subset \mathbb{R}^m$ can be factorized into scopes. We propose causal factored UCBVI (CF-UCBVI) in Algorithm 4 (Section C) whose structure is similar to C-UCBVI. We mainly discuss their differences in this section.

CF-UCBVI builds on C-UCBVI in terms of incorporating causal graph with the UCB value iteration idea. It calls CF-UCB-Q-values (Algorithm 5 in Section C) which returns upper confidence bounds on the Q-values, however, we construct the UCB bonus terms differently. Since we have prior knowledge of states factorization scopes, Algorithm 5 no longer needs to directly estimate $\mathbb{P}(s'|s, \mathbf{z})$ by counts. Instead, at every episode $k$, we estimate the transitions in each scope $\mathbb{P}_i(s'[i]|s[I_i], \mathbf{z})$ by $\hat{\mathbb{P}}_{k,i}(s'[i]|s[I_i], \mathbf{z})$ (see full definition in Algorithm 5). Using these scope-wise estimates, $\mathbb{P}(s'|s, \mathbf{z})$ can be estimated by $\prod_{i=1}^m \hat{\mathbb{P}}_{k,i}(s'[i]|s[I_i], \mathbf{z})$. We calculate the confidence bonus terms for every visited $(s[I_i], \mathbf{z})$ pair according to Algorithm 6 (Section C). The remaining procedures are quite similar to C-UCBVI. We update UCBs on $q$-values computed by value iteration using an empirical Bellman operator and the confidence bonus terms. See details in Algorithm 5 (Section C).

**Theorem 5 (Regret of CF-UCBVI)** *With probability $\geq 1 - \delta$, the regret of CF-UCBVI is:*

$$R_K = \tilde{O}\left(H \sum_{i=1}^{m} \sqrt{S_i S[I_i] Z T}\right).$$

*We omit small order terms that do not depend on $T = KH$.*

In Theorem 5, the regret bound consists a reduced term $\sum_{i=1}^{m} \sqrt{S_i S[I_i]}$ involving scope-wise state space parameters. The bound reduces to (1) when $m = 1$, otherwise, it improves the regret exponentially when $\mathcal{S}$ is high-dimensional. We provide the proof in Section B.

## 6. Causal linear MDPs

In this section, we consider function approximations on causal MDP dynamics. In particular, we show that linear MDP algorithms for non-causal MDPs can be well-incorporated with causal MDPs.

In linear MDPs, $\mathbb{P}(s'|s, a)$ and $R(s, a)$ are modeled by two linear functions and their corresponding feature functions are assumed to be known (Jin et al., 2019). In causal MDPs, since we already know the identity of parent variables $\mathbf{Z}$ that directly affect the state transition and reward, it is natural to instead model $\mathbb{P}(s'|s, \mathbf{z})$ and $R(s, \mathbf{z})$ via linear functions. We formally present the definition below.

**Definition 6 (Causal linear MDP)** *A causal linear MDP is a causal MDP equipped with a feature map $\phi : \mathcal{S} \times \mathcal{Z} \to \mathbb{R}^d$, where there exists $d$ unknown measures $\mu = (\mu^{(1)}, \ldots, \mu^{(d)})$ over $\mathcal{S}$ and an unknown vector $\omega \in \mathbb{R}^d$, such that for any $(s, \mathbf{z}) \in \mathcal{S} \times \mathcal{Z}$, we have*

$$\mathbb{P}(s'|s, \mathbf{z}) = \langle \phi(s, \mathbf{z}), \mu(s') \rangle \text{ and } R(s, \mathbf{z}) = \langle \phi(s, \mathbf{z}), \omega \rangle.$$

*Without loss of generality, we assume $\|\phi(s, \mathbf{z})\| \leq 1$ for all $(s, \mathbf{z})$, and $\max\{\|\mu(\mathcal{S})\|, \|\omega\|\} \leq \sqrt{d}$.*

One can re-write the state transition probability and the reward function for every $(s, a) \in \mathcal{S} \times \mathcal{A}$ using above features and the unknown linear coefficients in below.

$$R(s, a) = \langle \sum_{\mathbf{z} \in \mathcal{Z}} P(\mathbf{z}|s, a)\phi(s, \mathbf{z}), \omega \rangle \text{ and } \mathbb{P}(s'|s, a) = \langle \sum_{\mathbf{z} \in \mathcal{Z}} P(\mathbf{z}|s, a)\phi(s, \mathbf{z}), \mu(s') \rangle.$$

To this point, we demonstrate that linearly modeling the state transition and reward functions using parent variables is a special case of standard linear MDPs where the feature vector for every $(s, a) \in \mathcal{S} \times \mathcal{A}$ is $\psi(s, a) \triangleq \sum_{\mathbf{z} \in \mathcal{Z}} P(\mathbf{z}|s, a)\phi(s, \mathbf{z})$. Thus, we can easily extend linear MDP algorithms to our causal linear MDP setting. For example, applying Least-Squares Value Iteration with UCB (Jin et al., 2019) algorithm with features $\psi(s, a)$, one can achieve $\tilde{O}(\sqrt{d^3 H^3 T})$ regret.

## 7. Experiments

In this section, we conduct several experiments to validate the theoretical findings of our causal approaches. We compare our causal algorithms C-UCBVI and CF-UCBVI with two standard non-causal MDP or factored MDP algorithms: UCBVI (Azar et al., 2017) and F-UCBVI (Tian et al., 2020). Throughout, we use a causal factored MDP environment that allows us to compare the performance of all four algorithms. The state space is consisted of $d_s$-dimensional binary vectors, i.e. $\mathcal{S} = \mathcal{S}_1 \times \ldots \times \mathcal{S}_{d_s}, \mathcal{S}_i = \{0, 1\}$. There are $n$ manipulable variables $X_1, \ldots, X_n$, taking values

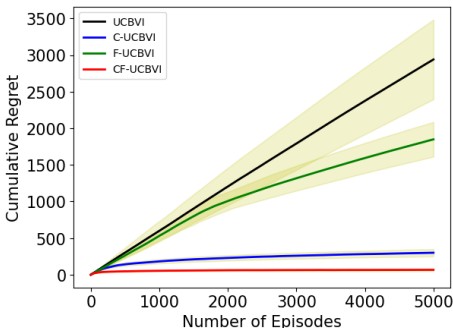

Figure 2: Cumulative regret v.s. number of episodes. $m = n = 3$, $H = 5$. We plot the averated cumulative regret in red, blue, green and black curves, and $1$-standard deviation for each method within the yellow shadow area.

from $\{1, \ldots, m\}$, $n$ non-manipulable parent variables of the reward and state variables $Z_1, \ldots, Z_n$, taking values from $\{1, 2\}$. The reward variable and the state transition variable directly depends on their parent variables $Z_1, \ldots, Z_n$. In each experiment below, we set $H$ differently and always guarantee that $Z = 2^n <= m^n = A$.

**Intervention set:** An intervention is denoted by $a = \text{do}(X_1 = i_1, \ldots, X_n = i_n)$, where $i_1, \ldots, i_n \in \{1, \ldots, m\}$. This means only non-parent variables can be intervened, while the parent variables of the reward are not under control.

**Reward generation.** We generate reward for every scope-wise state-$\mathbf{Z}$ pair $R(s_i, \mathbf{z})$ uniformly from $[0, 1]$. By factored MDP assumption, we calculate the state-$\mathbf{Z}$ pair rewards by $R(s, \mathbf{z}) = \sum_{i=1}^{d_s} R(s_i, \mathbf{z})$ and the state-action pair rewards by $R(s, a) = \sum_{\mathbf{z} \in \mathcal{Z}} R(s, \mathbf{z}) P(\mathbf{z}|s, a)$, where $P(\mathbf{z}|s, a)$ quantities are sampled from dirichlet distribution $\text{Dir}(\mathbf{1}_Z)$ for every $(s, a)$ pair.

**State transition.** We generate the scope-wise state-Pa-state transition probabilities from Dirichlet distribution $\text{Dir}(\mathbf{1}_2)$. By factored MDP assumption, we calculate the state-action-state transition probabilities by $\mathbb{P}(s'|s, \mathbf{z}) = \prod_{i=1}^{d_s} \mathbb{P}(s_i'|s_i, \mathbf{z})$. In this example, $I_i = \{i\}, i = 1, \ldots, d_s$.

**Experiment 1:** We begin with a simple case where $m = 4$ and $n = 3$. In this setting, we set the horizon $H = 5$, the dimension of state variable $d_s = 3$ and compare the performance of all four algorithms: UCBVI, C-UCBVI, F-UCBVI and CF-UCBVI over $K = 5000$ episodes. We repeat every algorithm for $10$ times and calculate the averaged regrets and their $1$-standard deviation confidence intervals at every episode. Regret comparison plot is displayed in Figure 2.

In this causal factored MDP environment, the regret plot shows that the only algorithm that uses causal knowledge and factored state space structure: CF-UCBVI outperforms other three algorithms while UCBVI has the highest regret. C-UCBVI and F-UCBVI use one of the structure properties, so their regret curves lie in the middle. It is hard to compare C-UCBVI and F-UCBVI. In general, when the causal relations are stronger than factored state structure relations, C-UCBVI outperforms F-UCBVI and vice versa. In this environment, it happens that C-UCBVI performs better.

**Experiment 2:** $m = 3, 4, 5, 6, 7; n = 3$. In this experiment, we fix $n = 3$ while changing the domain range of non-parent variables $m$ from 3 to 7. The number of interventions increases ex-

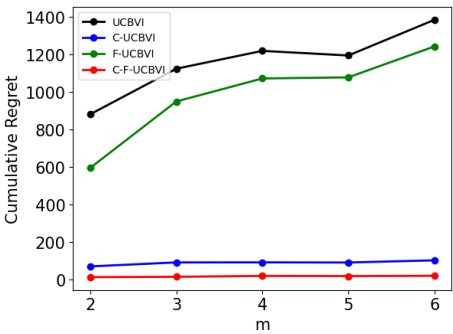
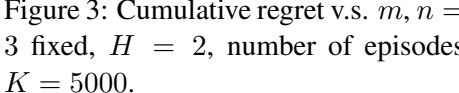
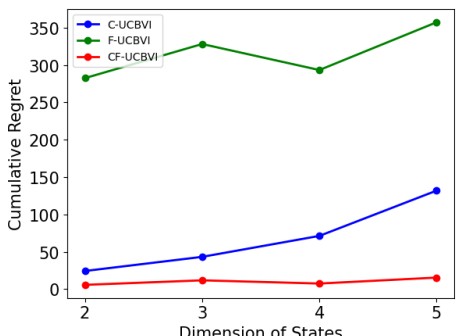

Figure 3: Cumulative regret v.s. $m$, $n = 3$ fixed, $H = 2$, number of episodes $K = 5000$.

Figure 4: Cumulative regret v.s. $d = 2, 3, 4, 5$, fix $m = 2, n = 3$, horizon $H = 2$, number of episodes $K = 5000$.

ponentially as $m$ increases, however, the number of parent variables value assignments $Z$ does not vary. For each algorithm, the cumulative regret after $K = 5000$ episodes is averaged over 10 simulations. Regret comparison plot is displayed in Figure 3.

As we increase the number of interventions, the regret curves show that the performances of C-UCBVI and CF-UCBVI are stable. The other two algorithms incur higher regrets when the intervention size becomes bigger. At every fixed $m$ value, the performance rank is the same as Figure 2.

**Experiment 3: vary state dimension** $d_s = 2, 3, 4, 5$. In this section, we fix $m = 3$ and $n = 3$ and compare three algorithms: C-UCBVI, F-UCBVI and CF-UCBVI across different state dimension settings: $d_s = 2, 3, 4, 5$. We set $K = 5000$ and repeat every algorithm for 10 times and compute the final averaged regret. We do not plot the regret curve for UCBVI because it does not converge until the end of 5000 episodes and thus the cumulative regret v.s. state dimension curve for UCBVI cannot reflect the true relation between $d_s$ and the regret of UCBVI. We can observe this phenomenon in Figure 2 where $d_s$ is only 2 and the UCBVI curve (in black) is almost straight up to $K = 5000$. Since we increase $d_s$ from 2 to 5 in this experiment, UCBVI converges even slower. Thus, we present the regret comparison among remaining three algorithms in Figure 4.

We observe that the regrets of F-UCBVI and CF-UCBVI algorithm do not vary too much, however, the regret of C-UCBVI increases significantly as $d_s$ increases. This phenomenon matches with our theories. C-UCBVI is the only algorithm out of the three who does not exploit the factored MDP environment, so its performance is the most sensitive to $d_s$. Due to the ignorance of causal knowledge in F-UCBVI, the regret curve of F-UCBVI stays at a higher value comparing to the other two methods.

## 8. Discussion

In this paper, we studied the causal (factored) MDPs. We proposed C-UCBVI and CF-UCBVI algorithms for the causal and causal factored MDP settings. Their regret bounds offer potentially exponential improvements over that of standard RL algorithms. In addition, we extended the causal MDP problem to its linear MDP variation.

There are several interesting directions we left for future work. First, we note that our approach can be easily adapted to an action hybrid setting, where some actions $\mathcal{A}_1$ lead to factored structure together with states and others $\mathcal{A}_2$ do not factorize with states, instead form a causal graph with small cardinality of total combinations for *key* or parent variables. One can combine F-UCBVI and CF-UCBVI by separately estimating the two types of state transition probabilities $\mathbb{P}(s'|s, a)$ where $a \in \mathcal{A}_1$ and $a \in \mathcal{A}_2$ using factored MDP techniques and our causal approach. Secondly, when the prior knowledge expressed in causal graphs and other causal quantities is mildly violated, a sensitivity analysis for our results can provide useful guidance to practitioners. Finally, our causal algorithms need background knowledge of certain conditional probabilities associated with the causal graphs. It will be promising to develop a causal algorithm that can learn the causal information and the MDP environment simultaneously and achieve lower regret than standard non-causal RL algorithms.

## Acknowledgments

AT acknowledges the support of NSF (via grant IIS-2007055) and Adobe (via a Data Science Research Award). We thank CLeaR 2022 reviewers for their numerous suggestions to improve the clarity of our paper.

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
