# OpenReview forum: "Efficient Reinforcement Learning with Prior Causal Knowledge"
_cclear.cc/CLeaR/2022/Conference — CLeaR 2022 Poster_

### Official Review · Reviewer_qWfE · 2021-11-21

**Confidence:** 4
**Overall Score:** 7

**Main Review:**

Quality: Overall this is a work of good quality. The motivating idea is interesting, and the solution is explained well and conveyed to the reader with precision.
Clarity: The work is generally well presented. The problem is motivated well, and the reader is introduced to the necessary notation in advance of reading the theorems. Furthermore, the use of diagrams to illustrate the problem is appreciated.
Originality: I have encountered some work which attempts to fuse causal graphical models with MDPs (e.g. Path Dependent Structural Equation Models by Srinivasan et al. 2020), so I would not call this a completely novel take on the problem. However, to my knowledge I have not encountered an approach that seeks to exploit structure using the Z variables in the manner that this paper discusses.
Significance: The work appears to be moderately significant, although as I discuss in the cons, I am not completely convinced of its practical utility.
Pros: The authors provide novel bounds on the regret of the policy by exploiting the structure afforded by Z. The authors provide a causal UCB value iteration algorithm and extend this to the factored MDP setting. Provided we believe the setup of the problem, these results drastically improve performance and should prove useful to analysts in complex settings such as healthcare.
Cons: My main concerns are about the quantity Z. In figure 1 there are graphs with e.g. $R$, the parent variables $Z^R$, and the intervention  variables $X^I$. What happens if there are edges $X^I \to R$?
Whilst I think that in many cases such an assumption is justifiable and indeed reasonable, this paper does not go far enough in checking the implications of making such an assumption. For instance, do promotional offers only affect purchase through exactly the three variables listed? How can we know if there aren’t other pathways, such as a person’s personal opinion about the brand (which might not be an observed quantity)?
It seems to me that this is a testable CI assumption that is being made here, and if it is violated then the factorizations in the second/third last equations of p. 6 are violated. As far as I can tell this is also assumed in the causal factor MDP, so all the presented algorithms hinge upon this testable assumption being true
What method do you propose to check this? Will you also consider sensitivity analysis to the violation of this assumption? In practice, if there are mild violations, how biased are the results? What happens to the bounds of these algorithms if such edges exist?
Answers to some of these questions would help a potential analyst figure out if their parent variables Z are suitable for the task.


**Summary:**

This paper introduces causal MDPs, which uses causal structures over both state and reward graphs. The motivation is that in practice many problems have causal structures which can be exploited to reduce the complexity of the relationship between the interventions and the states or rewards. This is done by assuming that a set of variables Z exists, such that the interventions A affect the reward only through these variables (and similarly for the state transitions). This leads to improved bounds on a variety of MDP algorithms since the relevant distributions now scale with respect to Z, not A.

---

> ### Author Response · Authors · 2021-12-01
> **Response to Reviewer qWfE**
>
> We thank the reviewer for your valuable comments and suggestions and we are glad you liked our paper. We address your questions below.
>
> 1. “I have encountered some work which attempts to fuse causal graphical models with MDPs (e.g. Path Dependent Structural Equation Models by Srinivasan et al. 2020)”: Thank you for pointing us to this related paper, we will cite and discuss it.
>
> 2. “What happens if there are edges X^I\rightarrow R?”: In this scenario, then $X^I$ are parents of $R$. In the causal MDP paragraph on page 5, we said that parent variables cannot be intervened, because otherwise one can simply restrict the intervention set to those only intervening over the parent variables set (leading to a trivial reduction to a standard MDP problem, see footnote 2) and a standard algorithm would work.
>
> 3. About the correctness of the causal graph (or testable conditional independencies assumption): We agree with the reviewer that the causal graph may not be correct or complete sometimes and it’s crucial to test the conditional independencies beforehand in that case. In this work, our main focus is on how to make use of side causal knowledge to speed up the learning process when the prior causal information is correct. For an end-to-end study where one tries to discover a set of Zs that can d-separate actions with R and actions with S, we recommend conducting conditional independence tests using observational data for subsets of variables that can potentially work as Zs.
>
> 4. “Will you also consider sensitivity analysis to the violation of this assumption? In practice, if there are mild violations, how biased are the results? What happens to the bounds of these algorithms if such edges exist?”: Thanks for your suggestion, sensitivity analysis to the violation of conditional independence assumption is a meaningful next step. By defining a proper causal graph misspecification criterion, similar to other model misspecification analysis, we expect that the regret bound should be inflated by an additive factor of \tilde{O}(\eps T) given \eps as the misspecification degree.

---

### Official Review · Reviewer_18wK · 2021-11-21

**Confidence:** 4
**Overall Score:** 6

**Main Review:**

The authors propose modifications to the existing UCB-type RL algorithms that rely on causal (actually more so probabilistic since they seem to have mainly utilized conditional independence) knowledge available a priori to achieve better regret than those that do not utilize such information. I liked the paper and the results make sense. But the proofs are missing and I cannot recommend acceptance without complete proofs. I would, however, be happy to change my score based on the authors' input to some clarification questions I have.


My detailed comments are as follows:
-------------------------------------------

"Otherwise, one can simply restrict the intervention set to those only intervening over PaR ∪ PaS, then the problem is trivially reduced to a standard MDP problem."
I believe a better way to handle this case is to present this as the solution first and then move onto the case when these cannot be intervened by - possibly by giving realistic examples of when that might happen - and then move onto the actual contribution.

The notion of "graphs at each state" is a bit confusing to me. Figure 1 does not really help clarify this. Is it possible to add an example graph that explicitly shows everything, time steps, states and graphs?

Could you comment on the assumption of having access to p(z|s,a)? Drawing connections to classical RL papers that make the same assumption might help.

"P(s'|s,a)"
In the factorization, the assumption S'\indep A|S,Z is used. Could you explain, justify this assumption? Hard to see without an explicit graph. Similarly, please explain R(s,a).

Along the same lines as the above, it is very hard to parse how strong factored MDP and factored reward definitons/assumptions are without seeing a causal graph structure.

"We note that extending RL algorithms from known rewards to unknown stochastic rewards poses no real difficulty (Azar et al., 2017)."
Could you elaborate on this statement? What is the high-level methodology for that? What do we lose in that case?

Unfortunately, I believe the authors have forgotten to add the appendix since Sections A and B are often cited but are not included in the pdf. Therefore, proofs are currently missing.

If I am understanding correctly, the proposed UCB-type algorithm changes exploration of S\times A to exploration of S\times Z since Z determines everything - being the parents of the quantities of interest. Then it is expected that when Z is much smaller, regret will be lower accordingly, which explains the form of regret in Theorem 4. Still, other terms also are different: for example we have S instead of \sqrt{S}, why is that? In general, seeing the proof will definitely help. But I also think it would be good to comment on why these other dependencies are changing for your algorithm.\

Causal factored-UCBVI seems to be utilizing the additional knowledge that states can be factorized into singletons (this part is not very clear to me since the definition was hard to parse - a graph example there will help here too). Again the reduction in regret with this side information is expected.

I think it will be good to make a clear point on exactly where causal knowledge is used. The fact that actions and their effects are the key to the RL settings makes me think that causality must have been used crucially but I was mostly able to identify the probabilistic information (conditional independence). Please point out exactly where ideas from causality is used. For example, what will fail if the given graph is not causal but only a Bayesian network? A walk-through over an example graph would be very helpful.

I am not sure what the conclusion is in Section 6. Is a theorem missing here? "Thus, we can easily extend linear MDP algorithms to" Perhaps authors didn't add a result since maybe they think it is trivial but I think adding an explicit result (even with a citation) would make the paper self-contained.

AFTER THE REBUTTAL

I am increasing my score based on the proofs and the comments of authors. My main comment is to really clarify where causality is used in the method, i.e., it is only used at the observational level to derive independence statements and use them in the algorithm.

**Summary:**

Official review of Reviewer 18wK

---

> ### Author Response · Authors · 2021-12-01
> **Response to Reviewer 18wK**
>
> We thank the reviewer for your valuable comments and we are glad you liked our paper. We address your questions below.
>
> 1. “The proofs are missing.”: We did include our proofs in our supplement materials (see clear2022.pdf in the zip file). We will appreciate it if you can check out our proofs in the appendix (Sec. A and Sec. B).
>
> 2. “a better way to handle this case is to present this as the solution first…”: Thank you for helping us improve the writing quality, we are willing to change this as you suggested.
>
> 3. “The notion of “the graph at every state” is a bit confusing…”: At each state, we use two causal graphs to characterize the reward generation and state transition (as Figure 1). We use grey arrows between layers to summarize the causal relations among those variables. What we meant by “the graph at every state” is that the underlying causal relations (captured by grey arrows) can change under different states, while the identity of those variables is unchanged (o.w. we can always include all variables to the causal graph). We take the reward generation graph as an example. For a loyal customer, a 10% price increase may not change the purchase behavior much; however, a new customer may easily go for another brand. So the orange variables affect the green variable differently for people in different states. We will add time steps and current state to Figure 1.
>
> 4. “Assumption of having p(z|s,a)”: Our work is an extension from causal bandits (e.g. [1][2][3]), where they focus on a bandit setup and do not have state variables. Their works assume knowledge of p(z|a) quantities are given. In our MDP setting, since state variables are also involved, we require access to p(z|s,a) instead of p(z|a). We will clarify this in our final version.
>
> 5. About known reward v.s. unknown stochastic reward: We take the C-UCBVI algorithm as an example. For stochastic unknown reward, one can simply replace the reward term R(s,z) in line 11 (Alg. 2) to the corresponding empirical reward estimations. All other steps remain unchanged and the resulting regret bounds are of the same order as our current results.
>
> 6. “other terms also are different: for example we have S instead of \sqrt{S}...”: Initially we used the proof techniques in Azar et al. (2017) and tried to bound $(\hat{P}-P^*)(\hat{V}-V^*)$ using Bernstein inequalities. There is a term in its bound: $S\sum_{i=1}^K\sum_{j=1}^H\sum_{\mathbf{z}}P(\mathbf{z}|s_{ij},a_{ij})\times \frac{1}{N_i(s_{ij},\mathbf{z})}$ (1)
> Due to the probability terms in (1), we need concentration inequalities to bound (1) by $\tilde{O}(S\sqrt{T})$. Thus, $S$ is in the final regret. This is different from the non-causal setting, where the above term is $S\sum_{i=1}^K\sum_{j=1}^H\frac{1}{N_i(s_{ij},a_{ij})} = O(S\log T)$ and $S$ isn't in the final $\tilde{O}(\cdot)$ regret due to the $\log T$ term. Since using Bernstein will still cause an $S$ in regret, we went for an easier approach to directly bound the transition model that also incurs $S$ in the regret bounds. See details in Sec. A and B in the appendix of clear2022.pdf (supplementary .zip file).
>
> 7. Towards understanding factored MDPs: Factored MDPs have been widely studied previously (e.g. [4][5]). We add causal structures over them. We will clarify this.
>
> 8. “it will be good to make a clear point on exactly where causal knowledge is used…...”:  We follow the terminology from existing works’ on causal bandits ([1][2][3]), where the action set is also composed of interventions on variables of a causal graph but they focused on a bandit setup. In these works, including our causal MDPs, interventions are defined via do-operator and causal graphs are used to model the relations among the intervenable variables, reward, state and other related variables. Causality is used, since the domain is modeled using causal graphs and our actions are interventions. Similar to causal bandit works, our main methodology is to make use of the conditional independencies between A, Z and R/S to speed up the learning process. If the reviewer has strong objections to the title of our work, we are also willing to change our title to something like: efficient reinforcement learning algorithms with prior causal information etc. We are also willing to add more discussion on how exactly causality is used in the paper (viz. mostly in the modeling not in the methodology).
>
> 9. “what the conclusion is in Section 6. Is a theorem missing here?”: No, there is no theorem in Sec. 6. We wanted to show that if we can model P(s’|s,z) and R(s,z) via linear functions, then existing linear MDP algorithm such as LSVI can be directly applied since we can then write P(s’|s,a) and R(s,a) via linear functions (two equations under Def. 6) and LSVI will work. We will add more explanations!
>
> [1] Lattimore et al. (2016)
>
> [2] Lu et al. (2020)
>
> [3] Nair et al. (2021)
>
> [4] Osband et al. (2014)
>
> [5] Xu et al. (2020)
>
> See titles for above papers in our paper references.

---

### Official Review · Reviewer_ex9n · 2021-11-26

**Confidence:** 3
**Overall Score:** 6

**Main Review:**

Originality: The authors have used simple assumptions like conditional independence and factored representations to gain on regret bounds. They make comparisons to prior work in causal reinforcement learning [Zhang et al.]. The work is closely linked to causal DBNs [Blondel et al., International Journal of Data Science and Analytics 2017] and path dependent causal models [Srinivasan et al. UAI 2021]: these papers introduce the concept of interventions in time-dependent systems, whose evolution changes with actions/interventions along the way. They also address confounding and effects on a system’s evolution upon intervention. Citing these would be beneficial to the scholarship of this work.

Significance: The paper claims to introduce causal MDPs. It would be important to understand the authors’ view on two issues:
(1) The usage of the word causality in this paper: The methodology in the paper does not seem to make use of counterfactual outcomes or the do-operator and the framework of causality. The main argument is that there is directionality in the dependence between A, Z and R/S. But does this extend beyond simple conditional independence, and is causality required for the arguments? Is it essential for nomenclature?
(2) Unobserved Confounding: The authors state they do not deal with unobserved confounding in this paper. However, the main draw of causal methods is their ability to deal with confounding and bias. It is not clear why this strong assumption about lack of confounding was made in the paper.

Technical quality: The methods introduced are conceptually fairly straightforward. The bounds can be valuable if the settings of use obey the assumptions in the paper. The experiments are simple, and the paper could have benefitted from a real-data example to show the true power of the method, and validity of assumptions.

Clarity: The paper is clearly written and easy to follow. It would be prudent to highlight the trail of assumptions that lead to the final results as they are currently sprinkled all over the writing, making it hard to reconcile the claims in the abstract with the actual ability of the method to deal with large cardinality of A and S. For eg, an assumption is that the states all have the same structure. But this isn’t evident from the writing in the rest of the paper.
The examples can be better structured to convey ideas more effectively. The paper could have benefitted from an experiment that retains any of the motivating example scenarios introduced in the paper.


**Summary:**

The authors introduce methods in reinforcement learning to deal with action and state spaces that have a large cardinality. For large action spaces, they propose Causal Markov Decision Processes (C-MDP), an extension of MDPs that incorporate a lower-dimensional set of variables Z that create a conditional independence between intervenable actions A and rewards R/states S. To deal with large state (and action) spaces they propose Causal Factored MDPs (CF-MDPs) that, like factored MDPs, have a factorizable state space. They propose C-UCBVI and CF-UCBVI algorithms and show that regret bounds for these cases are better than conventional bounds.

---

> ### Author Response · Authors · 2021-12-01
> **Response to Reviewer ex9n**
>
> We thank the reviewer for your valuable comments! We address your concerns below:
>
> 1. Originality: We thank the reviewer for pointing us to the two papers that have introduced the concept of interventions in time-dependent systems. We will cite them in our final version.
>
> 2. Significance:
>
> a) “The usage of the word causality in this paper…”: We follow the terminology from existing works’ on causal bandits ([1][2][3]), where the action set is also composed of interventions on variables of a causal graph but they focused on a bandit setup (while ours is MDP). In these works, including our causal MDPs, interventions are defined via do-operator (see page 5) and causal graphs are used to model the relations among the intervenable variables, reward, state and other related variables. Causality is used, since the domain is modeled as causal graphs and our actions are interventions. Similar to causal bandit works, our main methodology is to make use of the conditional independencies between A, Z and R/S to speed up the learning process. If the reviewer has strong objections to the title of our work, we are also willing to change our title to something like: efficient reinforcement learning algorithms with prior causal information etc. We are also willing to add more discussion on how exactly causality is used in the paper (viz. mostly in the modeling not in the methodology).
>
> b) About unobserved confounding (UC): We agree that the non-existence of UC assumption is strong in some sense. The reason why we made this assumption is that UCs can cause trouble for both non-stationary optimal policies for RL and causal model violation, which overall is a very complicated scenario outside of our current scope. We hope our work can be a nice starting point for future works when UCs exist.
>
> 3. "Technical quality: ......The experiments are simple......": Our paper is motivated by real applications, but our main focus is on the theoretical side. Applying our method to real examples will require more work such as obtaining the domain causal knowledge and will be a good next step!
>
> 4. Clarity: Thank you for helping us improve the writing.
>
> a) We will make the experiment section more well-organized in our final version.
>
> b) “an assumption is that the states all have the same structure”: We do not need all states having the same structure. We set the states to have the same structure just for simplicity.
>
> [1] Lattimore et al. (2016) Causal Bandits: Learning Good Interventions via Causal Inference
>
> [2] Lu et al. (2020) Regret Analysis of Bandit Problems with Causal Background Knowledge
>
> [3] Nair et al. (2021) Budgeted and Non-budgeted Causal Bandits

---

### Official Review · Reviewer_ULh7 · 2021-11-27

**Confidence:** 2
**Overall Score:** 9

**Main Review:**

Although I am far from an expert in this area, this appears to be a highly innovative paper.

The authors first introduce a new type of decision problem -- causal Markov decision problems CMDPs.  A CMDP is a special case of a Markov decision problems (MDP), but reasonable performance in MDPs is typically achieved by assuming either (i) the state and action spaces are both finite and relatively small, OR (ii)  the state and action space can be factored in  a very particular way.  The authors note that, in certain settings (like healthcare and digital marketing), neither of those assumptions is realistic.  The authors then introduce CMDPs to model decision problems like those encountered in healthcare and marketing.  They briefly argue that, in such situations, a different type of factorization condition is plausible if one assumes the underlying actions, state, and reward structures  reflect some underlying causal  structure.  The authors then generalize UCBVI algorithms to CMDPs, prove a regret bound, and analyze the performance of their algorithms on simulated data.

The paper is clearly written and well-organized.  Assuming the details of the arguments are correct, I think the only major weakness of the paper is that the authors do not apply their algorithms to real data from the healthcare and marketing settings the purportedly motivated this theoretical work.  Perhaps the application is trivial, but as a non-expert, it's not immediately clear  to me if the assumption used to derive the regret bounds is plausible in those settings.  Nonetheless, considered as a theoretical paper, this seems strong to me.

**Summary:**

The authors introduce a new type of learning problem called CMDPs, and they generalize UCBVI algorithms to CMDPs.

---

> ### Author Response · Authors · 2021-12-01
> **Response to Reviewer ULh7**
>
> We thank the reviewer for appreciating our paper! We are glad that you find our paper innovative, clear and theoretically strong. Our paper is motivated by real applications, but our main focus is on the theoretical side. Applying our method to real examples will require more work such as obtaining the domain causal knowledge and will be a good next step!

---

### Decision · Program_Chairs · 2022-01-12

**Decision:**

Accept (Poster)

**Comment:**

In this paper, the authors aim to generalize MDPs to allow modeling causal relationships within each state, by using ideas from the causal bandits literature. The authors use this factored representation to derive regret bounds that depend do not necessarily depend on the number of interventions and states.

It seems that the causal MDP representation isn't novel, but is a special case of existing approaches that generalize MDPs to allow states to be represented by causal graphs (possibly with unobserved confounding), see e.g. comments of reviewer ex9n. That said, the regret bound results appear to me to be novel.

Overall the reviewers received the paper positively, and the paper ought to be published. However, the authors should be clear about relationship of their proposal to prior work.